

# Double decoupling effectiveness of water consumption and wastewater discharge in China's textile industry based on water footprint theory

Yi Li[1] and Yi Wang[2]

[1] School of Tourism and Resource Environment, Qiannan Normal University for Nationalities, Duyun, China
[2] School of Economics and Management, Zhejiang Sci-Tech University, Hangzhou, China

## ABSTRACT

As a traditional pillar industry in China, the textile industry has been intensifying the pressure of the water resource load and its reduction of water environment emissions over the years. Decoupling water resource consumption and wastewater discharge require decoupling from economic growth to realise the sustainable development of the textile industry. On the basis of water footprint and decoupling theories, this paper analysed the water consumption decoupling, wastewater discharge decoupling, as well as the double decoupling of water consumption and wastewater discharge of China's textile industry and its three sub-industries (Manufacture of Textile, Manufacture of Textile Wearing and Apparel, Manufacture of Chemistry) from 2001 to 2015. In those years, the sum of the decoupling index in the double-decoupling years is 249, lower than that in high-decoupling years of water consumption (250) and wastewater discharge (325). Compared with the decoupling of water consumption and of wastewater discharge, the double decoupling is lower, which proves that the conditions for realizing double decoupling are stricter. The double decoupling analysis of water consumption and wastewater discharge, namely, the overall consideration of water resource consumption and water environment pollution, could be used to more effectively promote the realisation of water decoupling in the textile industry.

# BACKGROUND

Water resources per capita in China is 2,710 m$^3$, which is less than 1/4 of the world's per capita water resources, ranking 88th in the world. China's water shortage has become a major obstacle to economic and social development (*Lv, 2017*). At the same time, China's industrial wastewater discharge is relatively large, and the implementation of wastewater treatment measures is not in place, resulting in many wastewaters not meeting the professional discharge standards directly into other water bodies, which has brought great pollution to the environment (*Wang & Chen, 2016*). In 2015, China's total water consumption was 610.32 billion m$^3$, where industrial water consumption accounted

Corresponding author
Yi Li, liyihangzhou@qq.com

for 21.9% (*Ministry of Water Resources of the People's Republic of China, 2015*), with 73.53 billion tons of wastewater, 27.1% of industrial, 22.24 million tons (Mt) of chemical oxygen demand in wastewater and 13.2% in industrial wastewater (*Ministry of Environmental Protection of the People's Republic of China, 2015*).

The textile industry is a traditional pillar industry in China. It has a comparative advantage and an increasing international competitiveness in the international market. China is the world's leading textile producer and exporter. In 2017, China's textile industry exports amounted to 27.455 billion US dollars, accounting for 36.8% of the world's total textile exports (*China National Textile and Apparel Council, 2018*). With the rapid development of China's textile industry, the problems of high energy consumption, high water consumption and high emissions seriously restrict the sustainable development of China's textile industry's economy and ecological environment (*Wang et al., 2012*; *Yu et al., 2005*; *Li et al., 2017c*). In 2015, textile industry wastewater discharged 1.84 billion tons, ranking third among 41 key industries in China for five consecutive years (2011–2015). In 2015, the textile industry emitted 206 thousand tons of chemical oxygen demand, ranking fourth among 41 key industries in China (*Ministry of Environmental Protection of the People's Republic of China, 2011a*, *2011b*, *2012a*, *2012b*, *2013a*, *2013b*, *2014a*, *2014b*). At the same time, the recycling level of water resources in China's textile industry is on the low side. The water reuse rate in the textile industry is less than 70%, which is lower than the average level of 80% in the national industry. The water reuse rate in printing and dyeing industry is only 30% (*China National Textile and Apparel Council, 2018*).

Therefore, giving full play to the leading role of the textile industry in China's national economy and realizing the green development and sustainable development of textile industry play an important role in constructing Chinese ecological civilization, promoting the development of related industries and stimulating the growth of domestic demand. Environmental protection is not only the responsibility of national environmental management (*Tan & Fang, 2016*), but also the responsibility of energy conservation and emission reduction of related industries. According to the textile industry development plan 2016–2020, by 2020, the water intake per textile unit of industrial added value will decrease by 23%, and the total discharge of major pollutants will decrease by 10%. The sustainable development of China's textile industry is facing the dual dilemma of serious water shortage and excessive wastewater discharge. Decoupling is the inevitable choice to break the coupling relationship of the Chinese textile industry between economic growth and water resources and environment.

## Literature

Decoupling theory is derived from physical concepts and refers to situations in which the relationship between two or more interrelated variables decreases or ceases to exist (*Ang, 2004*). The Organization for Economic Co-operation and Development (OECD) defines it as the weakening of the synchronous relationship between economic growth and environmental pressure and sets the ratio of pollution emissions in the final stage to GDP and in the base period as the 'decoupling index', which divided decoupling into absolute and relative (*Organization for Economic Co-operation and Development (OECD), 2002*).

*Tapio (2005)* put forward a decoupling elastic model and divided the decoupling relations into several types: weak decoupling, strong decoupling, weak negative decoupling, strong negative decoupling, expansive negative decoupling, recessive decoupling, expansive coupling and recessive coupling. The main methods of decoupling evaluation include a comprehensive analysis of variation, decoupling index, elastic analysis and decoupling analysis based on complete decomposition technology, IPAT model, descriptive statistical analysis, econometric analysis and differential regression coefficient (*Zhong et al., 2010*). Moreover, the coupling relationship between economic development and environmental pressure increases the difficulty of reducing global emissions, and the effective use of water resources is extremely important to promote economic development (*Azad, Ancev & Hernández-Sancho, 2015*). The decoupling method has been used extensively in research on water resources and environment, mostly from the perspectives of decoupling of water resource consumption and economic growth, as well as the decoupling of wastewater discharge and economic growth.

In the field of water consumption decoupling, *Lesin et al. (2017)* obtained the decoupling relationship between economic growth and water consumption by discussing the reasons why water pollution supports the weak regional economic growth. *Wang (2015)* applied decoupling theory and *Tapio (2005)* elasticity analysis to analyze the relationship between China's economic growth and water resource utilisation from the perspective of time and space and concluded that the decoupling between industrial production and industrial water use was weaker than that in agriculture. *Li et al. (2017b)* analysed the decoupling elasticity of water consumption and economic growth of the textile industry and its three sub-industries and concluded that the scale factor of the textile industry drove water resource consumption and that the efficiency factor inhibited the increase of total water resource consumption. In addition, the decoupling theory of water resources could also be used to predict the decoupling status of resources and economic development in the future. *Wu (2014)* predicted that China's economic development and water resources utilisation would be absolutely decoupled by 2020 through the establishment of a decoupling analysis model. *Gilmont (2015)* used the decoupling theory to discuss the decoupling relationship between the virtual water flow of international food trade and the tonnage of food imports and corroborated that trade reduced the blue water footprint. By discussing the decoupling practice between economic development and resource consumption, *Zhou, Li & Zhou (2014)* put forward policy suggestions focusing on the real economy for China to achieve sustainable and stable economic growth. Similarly, on the basis of decoupling theory, *Lesin et al. (2017)* innovated the management method of water resources in Russia.

In the field of wastewater discharge decoupling, *Wang (2013)* used the decoupling index to analyze the decoupling relationship between water environment pollution and economic growth in China's textile industry. The paper concluded that the grey water footprint of China's textile industry shows a trend of strong decoupling (*Wang, 2013*). *Li & Sun (2016)* analysed the decoupling status between water environmental pressure and economic growth in China from 1991 to 2013, taking industrial wastewater discharge

as the index of water environmental pressure. The paper affirmed that technological effect had no significant impact on environmental regulation and that the structural and scale effects had a significant impact (*Li & Sun, 2016*). *Zhang & Yang (2014)* showed a weak decoupling relationship existed between agricultural water environment and crop yield. *Li et al. (2018)* used the textile industry as an example to analyze the decoupling elasticity and contended that the scale factor was the largest contributor to promote wastewater discharge and that the efficiency and structure factors were the driving forces for wastewater reduction. *Yu et al. (2013)* combined the trend of China's ecological efficiency development and the reasons for its dynamic change. China's resource utilisation, energy consumption and wastewater discharge were relatively decoupled from its economic development from 1978 to 2010 and absolutely decoupled from economic growth in terms of smoke and dust emission, chemical oxygen demand and ammonia nitrogen. Moreover, on the basis of the wastewater discharge decoupling study, reducing the discharge of industrial wastewater is the key to reducing the industrial grey water footprint, protecting the water environment and building the sustainable development of the industrial economy (*Ban, Zhang & Cao, 2017*).

In actual industrial production and the ecological economic system, water consumption, wastewater emission and economic development comprise a whole system. Only one aspect of decoupling between water resource consumption and economic growth, or between water environment discharge and economic growth, would neglect the systematic integrity of resources, environment and economy, and the results would not be comprehensive. The premise of realizing sustainable industry development is the decoupling of economic growth from water resource consumption and from wastewater discharge, that is, the double decoupling of water consumption and wastewater discharge (*Li et al., 2017a*). *Lu, Wang & Yue (2011)* deduced resource decoupling and emission decoupling indexes from the IGT and $I_eGTX$ equations, respectively. According to the decoupling index, the decoupling degree of resource consumption, waste emissions, and Gross Domestic Product is divided into three levels: absolute decoupling, relative decoupling and undecoupling. Through empirical analysis, developing countries with rapid economic growth were concluded to be less likely to obtain a higher decoupling index than developed countries with slower economic growth. *Gai, Hu & Ke (2013)* analysed the decoupling status of the resource consumption, environmental pollution and economic development of 16 cities in the Yangtze River Delta region from 2000 to 2009 and found differences in time as well as indicators between resource consumption, environmental pressure and the decoupling degree of comprehensive resource and environment. Another paper calculated the blue water, grey water and water footprints of China's textile industry from 2001 to 2014 but mainly analysed the factors and contributions that affected the water footprint (*Li et al., 2017a*). The effectiveness of single decoupling of water resource consumption and wastewater discharge, as well as double decoupling of water consumption and wastewater discharge, were not compared.

On the basis of the above research, the development of the textile industry is restricted by water resources and the water environment. Research on single water consumption

decoupling or single wastewater discharge decoupling showed defects. Thus, the double decoupling problem of water consumption and wastewater discharge requires further attention. We introduce decoupling indices, assign a value to the degree of decoupling, and compare their effectiveness (*Guo & Zhang, 2013*). Taking China's textile industry as an example, the present study further explores the relationship between the double decoupling of water consumption and wastewater discharge and the single decoupling of the water consumption or wastewater discharge. Water footprint was used to characterise the water resources environment, and the decoupling relationship between water resources environment and economic growth was obtained by using the *Tapio (2005)* elastic analysis method. For further exploration of the internal mechanism of water decoupling and putting forward countermeasures and suggestions, the validity of the results of water consumption decoupling (blue water footprint decoupling), wastewater discharge decoupling (grey water footprint decoupling), and double decoupling of water consumption and wastewater discharge (water footprint decoupling) were analysed and compared by using the assignment partition method.

## METHODS AND DATA

### Water footprint accounting method of the textile industry

Water footprint includes blue water, green water and grey water footprints. In the textile industry production, blue water footprint refers to the consumed surface water or groundwater resources. Green water footprint refers to the total amount of consumed water from rainwater, soil and air. Given its small consumption amount in China's textile industry, green water footprint is excluded in this research. Grey water footprint refers to the amount of water needed to accommodate the pollutants produced in the production process. Blue water footprint calculation directly takes the textile industry water resource consumption. Grey water footprint accounting refers to the total amount of fresh water required for the absorption and assimilation of pollutants by natural background concentration and existing water quality environment before the treatment of textile industry wastewater.

The specific formulas are as follows:

$$\text{WF} = \text{WF}_b + \text{WF}_{gy} \tag{1}$$

$$\text{WF}_b = \text{W}_u \tag{2}$$

$$\text{WF}_{gy} = \max\left[\frac{L[k]}{C_s[k] - C_n[k]}\right] \tag{3}$$

In Eqs. (1), (2) and (3), WF is water footprint; $\text{WF}_b$ is blue water footprint; $\text{WF}_{gy}$ is grey water footprint and $\text{W}_u$ is the total water consumption of the textile industry. All units are in Mt. $L[k]$ is the amount of pollutant $k$ in the wastewater discharged by the textile industry; $Cs[k]$ is the concentration limit value of pollutant $k$ stipulated in the pollutant emission standard and $Cn[k]$ is the background concentration of pollutant $k$ in natural water body. These units are in Mt of pollutant/year, with $Cn[k]$ assumed to be zero Mt of pollutant/year.

**Table 1  Emission limits for wastewater pollutants in the textile industry.**

| Limits of pollutant species | Limits |
|---|---|
| pH | 6–9 |
| $COD_{Cr}$ | 100 |
| $BOD_5$ | 25 |
| Suspended matter | 60 |
| Chroma | 70 |
| Aniline | 1.0 |
| Total nitrogen | 20 |
| Total phosphorus | 1.0 |
| $ClO_2$ | 0.5 |
| Organic halogen | 15 |
| Sulfide | 1.0 |
| Ammonia nitrogen | 12 |
| Hexavalent chromium | 0.5 |

Table 1 shows the calculation formula of the water footprint. The calculation standard of the concentration limit of pollutant $k$ stipulated in the pollutant discharge standard adopts the relevant stipulation of *the water pollutant discharge limit value of existing enterprises in the Textile Dyeing and Finishing Industry* (GB13458-2013) (*Ministry of Environmental Protection of China, 2013*). The concentrations of chemical oxygen demand ($COD_{Cr}$), biological oxygen demand ($BOD_5$), suspended solids, ammonia nitrogen and other water pollutants in natural water are extremely low. In addition, the natural concentrations of these pollutants in different parts of the water body vary significantly. As such, collecting accurate data of these pollutant indicators is difficult. This paper assumes that $C_n[k]$ is 0, which means that the grey water footprint value will be slightly smaller. When calculating the grey water footprint of the textile industry, the value according to $COD_{Cr}$ is the largest.

## Decoupling theory

The decoupling accounting method adopted in this paper is based on the decoupling elasticity method proposed by *Tapio (2005)*. The main accounting index is the decoupling elasticity coefficient, which is defined as the ratio between resource consumption or environmental pressure change rate and economic change rate in a certain period of time. The formula is as follows:

$$D = \frac{\%\Delta VOL}{\%\Delta G} = \frac{(VOL_t - VOL_{t-1})/VOL_{t-1}}{(G_t - G_{t-1})/G_{t-1}} \quad (4)$$

In Eq. (4), $D$ is the elastic coefficient; $VOL_t$ and $VOL_{t-1}$ are the resource consumption or environmental pressure in the $t$ year and $t-1$ year, respectively; $G_t$ and $G_{t-1}$ are the GDP in the $t$ year and $t-1$ year, respectively; $\%\Delta VOL$ is the growth rate of resource consumption or environmental pressure and $\%\Delta G$ is the economic growth rate. The decoupling state is divided into eight types. Figure 1 shows that strong decoupling

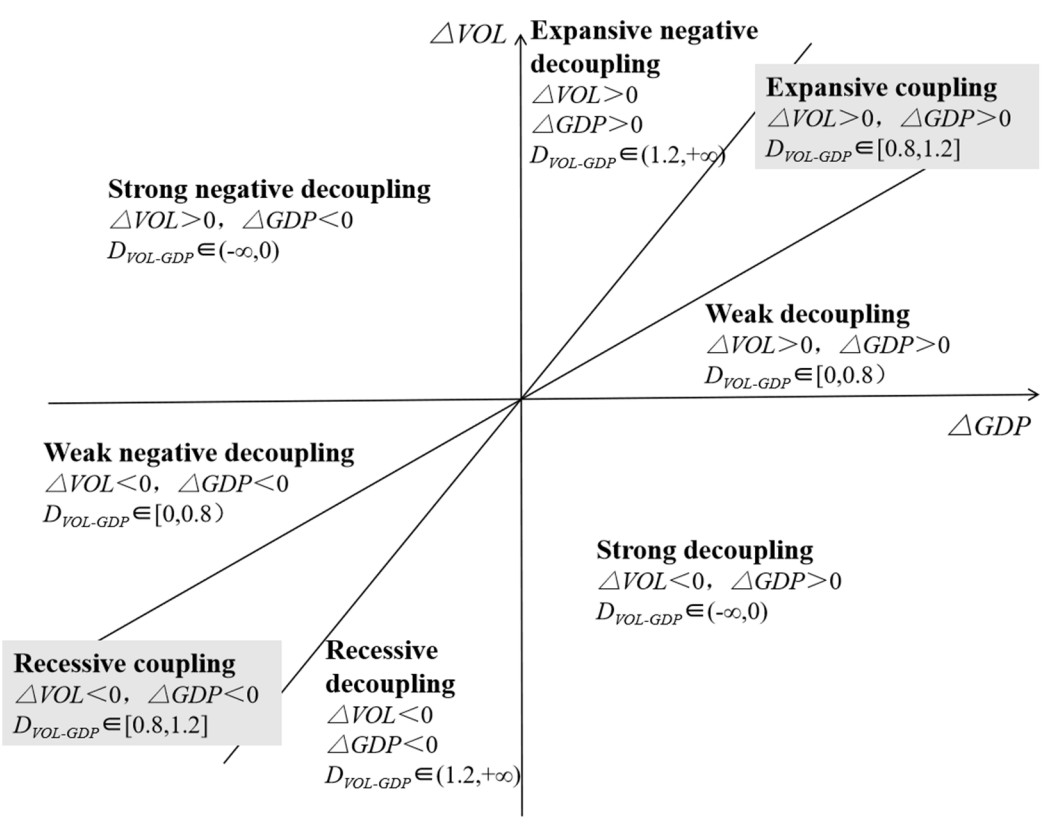

**Figure 1 Decoupling standard of decoupling elasticity method.** The decoupling state is divided into eight types. It shows that strong decoupling is the most ideal state of sustainable development, followed by weak decoupling. In addition, the other states are all not ideal, and the strong and negative decoupling is the most unsatisfactory state.       

is the most ideal state of sustainable development, followed by weak decoupling. In addition, the other states are all not ideal, and the strong and negative decoupling is the most unsatisfactory state.

### Improvement of decoupling model based on water footprint

The water footprint of the textile industry shows the double effect of water consumption and wastewater discharge. The double decoupling of these two factors is the decoupling relationship between the water footprint of the textile industry (WF) and the total output value of the textile industry ($G$). In this paper, the decoupling elasticity method is adopted to define $D_{G-WF}$ as the decoupling elasticity index between the water footprint and the economic growth of the textile industry. The decoupling state is calculated as the elasticity index. The formula is as follows:

$$D_{G-WF} = \frac{\%\Delta WF}{\%\Delta G} = \frac{\%(WF^t - WF^{t-1})G^{t-1}}{\%(G^t - G^{t-1})WF^{t-1}} \tag{5}$$

In Eq. (5), $WF^t$ and $WF^{t-1}$ represent the water footprint of the textile industry in $t$ year and $t-1$ year, respectively. The water footprint growth rate of the textile industry and the

growth rate of the total industrial output value of the textile industry can be obtained by calculating the corresponding data of the two time points.

### Assignment of decoupling degree

According to Tapio's (2005) deepening decoupling index, the decoupling status between water footprint and economic growth of the textile industry is summarised into eight types. The decoupling results of water consumption, wastewater discharge and the double decoupling of water consumption and wastewater discharge are analysed and compared to intuitively understand the change of the decoupling state. First, the decoupling states are divided into several intervals and then ranked according to the advantages and disadvantages of decoupling states. Finally, decoupling elastic coefficients of each interval are assigned from 1 to 28 (Guo & Zhang, 2013). The larger decoupling index shows better decoupling state between the water footprint and economic growth of the textile industry. Table 2 shows the divisions.

## Data

According to China's National Economic Industry Classification (GB/T 4754-2011) (National Bureau of Statistics of the People's Republic of China, 2011), the textile industry's gross industrial product and total wastewater discharge are regarded as the sum of data of three sub-industries: Manufacture of Textile; Manufacture of Textile Wearing and Apparel; and Manufacture of Chemical Fibers. With 2001–2014 as the research interval, the total output value of the textile industry (converted to constant price in 2014), total water consumption and pollutant content of wastewater used by the research institute are included in the China Environmental Yearbook (Ministry of Environmental Protection of the People's Republic of China, 2002–2006) and China Environmental Statistics Annual Report (Ministry of Environmental Protection of the People's Republic of China, 2006, 2007, 2008, 2009, 2010, 2011a, 2011b, 2012a, 2012b, 2013a, 2013b, 2014a, 2014b). However, given the misprints in the industrial water consumption data of MCF in 2012, this paper selected the average value of data in 2011 and 2013 as replacement.

# RESULTS AND DISCUSSION

## Results and analysis of water footprint decoupling in the textile industry

### Blue water footprint (water consumption) decoupling

Table 3 shows the calculation results of the decoupling elasticity of China's textile industry from 2002 to 2015. Except for 2002–2004 and 2014, the growth rate of water resource consumption in the textile industry is less than its economic growth rate. Except for 2012, the economic output of the textile industry showed positive growth in other years. In the sample period, the growth rate of water resources consumption decreased for 7 years.

Data Sources: $\%\Delta WC$ is the growth rate of water consumption; $\%\Delta G$ is the economic growth rate; and $D_{G-WC}$ is the decoupling elasticity of water consumption and economic growth.

**Table 2 Standard of double decoupling degree of water consumption and wastewater discharge in the textile industry.**

| Decoupling degree | Relationship between water footprint and economic growth | Evaluation value of decoupling status | |
|---|---|---|---|
| | | Elasticity coefficient | Decoupling index |
| Strong decoupling | $\Delta WF < 0, \Delta G > 0$ $D_{G-WF} \in (-\infty, 0)$ | $(-\infty, -0.6)$ | 28 |
| | | $[-0.6, -0.4)$ | 27 |
| | | $[-0.4, -0.2)$ | 26 |
| | | $[-0.2, 0)$ | 25 |
| Weak decoupling | $\Delta WF > 0, \Delta G > 0$ $D_{G-WF} \in [0, 0.8)$ | $[0, 0.2)$ | 24 |
| | | $[0.2, 0.4)$ | 23 |
| | | $[0.4, 0.6)$ | 22 |
| | | $[0.6, 0.8)$ | 21 |
| Recessive decoupling | $\Delta WF < 0, \Delta G < 0$ $D_{G-WF} \in (1.2, +\infty)$ | $(1.8, +\infty)$ | 20 |
| | | $(1.6, 1.8]$ | 19 |
| | | $(1.4, 1.6]$ | 18 |
| | | $(1.2, 1.4]$ | 17 |
| Expansive coupling | $\Delta WF > 0, \Delta G > 0$ $D_{G-WF} \in [0.8, 1.2]$ | $[0.8, 1.0)$ | 16 |
| | | $[1.0, 1.2)$ | 15 |
| Recessive coupling | $\Delta WF < 0, \Delta G < 0$ $D_{G-WF} \in [0.8, 1.2]$ | $[1.0, 1.2)$ | 14 |
| | | $[0.8, 1.0)$ | 13 |
| Expansive negative decoupling | $\Delta WF > 0, \Delta G > 0$ $D_{G-WF} \in (1.2, +\infty)$ | $(1.2, 1.4)$ | 12 |
| | | $(1.4, 1.6)$ | 11 |
| | | $(1.6, 1.8]$ | 10 |
| | | $(1.8, +\infty)$ | 9 |
| Weak negative decoupling | $\Delta WF < 0, \Delta G < 0,$ $D_{G-WF} \in [0, 0.8)$ | $[0.6, 0.8)$ | 8 |
| | | $[0.4, 0.6)$ | 7 |
| | | $[0.2, 0.4)$ | 6 |
| | | $[0, 0.2)$ | 5 |
| Strong negative decoupling | $\Delta WF > 0, \Delta G < 0$ $D_{G-WF} \in (-\infty, 0)$ | $[-0.2, 0)$ | 4 |
| | | $[-0.4, -0.2)$ | 3 |
| | | $[-0.6, -0.4)$ | 2 |
| | | $(-\infty, -0.6)$ | 1 |

Table 3 shows that the decoupling state during the Tenth Five-Year Plan period (2001–2005) is unstable but improved in the Eleventh Five-Year Plan period (2006–2010). During the Twelfth Five-Year Plan period (2011–2015), decoupling is relatively good, but unstable. In terms of strong decoupling, only the decoupling elasticity in 2003 is an absolute value at 1.32 whereas the other values are less than one. However, the degree of decoupling is weak and needs further improvement. In particular, the most unsatisfactory state of decoupling is in 2012 with a strong negative decoupling, which is closely related to the increase of water consumption and economic recession in that year. In 2014, the decoupling elasticity reached its maximum value of 13.47 in the sample period. The total industrial water consumption in that year increased considerably, resulting

**Table 3 Decoupling elasticity index of water resources consumption in China's textile industry from 2002 to 2015.**

| Year | %ΔWC | %ΔG | $D_{G-WC}$ | Degree of decoupling |
|------|------|------|------|------|
| 2002 | 14.23 | 7.15 | 1.99 | Expansive negative decoupling |
| 2003 | −10.11 | 7.66 | −1.32 | Strong decoupling |
| 2004 | 18.79 | 18.67 | 1.01 | Expansive coupling |
| 2005 | 15.01 | 22.86 | 0.66 | Weak decoupling |
| 2006 | −2.79 | 26.64 | −0.10 | Strong decoupling |
| 2007 | 10.28 | 18.85 | 0.55 | Weak decoupling |
| 2008 | −1.72 | 40.73 | −0.04 | Strong decoupling |
| 2009 | −0.34 | 6.38 | −0.05 | Strong decoupling |
| 2010 | 2.43 | 12.40 | 0.20 | Weak decoupling |
| 2011 | −16.59 | 25.39 | −0.65 | Strong decoupling |
| 2012 | 2.12 | −3.09 | −0.69 | Strong negative decoupling |
| 2013 | −2.56 | 4.84 | −0.53 | Strong decoupling |
| 2014 | 10.10 | 0.75 | 13.47 | Expansive negative decoupling |
| 2015 | −1.84 | 7.42 | −0.25 | Strong decoupling |

in a rebound between the economic development of the textile industry and resource consumption.

### Grey water footprint (wastewater discharge) decoupling

Table 4 shows the decoupling elastic calculation results of the grey water footprint of the textile industry and economic growth in China from 2002 to 2015.

Data Sources: $\%\Delta WF_g$ is the growth rate of wastewater discharge; $\%\Delta G$ is the economic growth rate; and $D_{G-WF_g}$ is the decoupling elasticity of wastewater discharge and economic growth.

From the overall decoupling trend, China's textile industry economic growth and grey water footprint shows a good decoupling. Apart from the negative economic growth rate of the textile industry in 2012 that led to the recession decoupling, the other years mainly showed strong or weak decoupling, with the overall trend showing an increasing decoupling. Specifically, the decoupling trend can be divided into three stages.

From 2002 to 2003, the grey water footprint and economic growth achieved strong decoupling for two consecutive years. The decoupling status was ideal. The textile industry's economic growth rate remained approximately 7%, whereas the reduction rate of grey water footprint exceeded 5%. From 2004 to 2007, the grey water footprint of the textile industry and economic growth showed a weak decoupling state for four consecutive years. Moreover, the gross industrial output value of China's textile industry continued growing at a relatively high speed. The growth rate of grey water footprint is also positive, which shows no absolute reduction at this stage. From 2008 to 2012, although the total industrial output value of the textile industry keeps rising, the growth rate of grey water footprint alternates positively and negatively, resulting in a strong and weak oscillating decoupling state. In 2012, a state of recession decoupling occurred as the grey water footprint of the textile industry achieved negative growth, and the economic

**Table 4 Decoupling elasticity of grey water footprint in China textile industry from 2002 to 2015.**

| Year | %ΔWF$_g$ | %ΔG | $D_{G–WFg}$ | Degree of decoupling |
|------|----------|-----|-------------|----------------------|
| 2002 | −5.99 | 7.15 | −0.84 | Strong decoupling |
| 2003 | −5.81 | 7.66 | −0.76 | Strong decoupling |
| 2004 | 13.58 | 18.67 | 0.73 | Weak decoupling |
| 2005 | 4.01 | 22.86 | 0.18 | Weak decoupling |
| 2006 | 5.55 | 26.64 | 0.21 | Weak decoupling |
| 2007 | 3.04 | 18.85 | 0.16 | Weak decoupling |
| 2008 | −5.90 | 40.73 | −0.14 | Strong decoupling |
| 2009 | 4.13 | 6.38 | 0.65 | Weak decoupling |
| 2010 | −2.90 | 12.40 | −0.23 | Strong decoupling |
| 2011 | 5.13 | 25.39 | 0.20 | Weak decoupling |
| 2012 | −4.56 | −3.09 | 1.48 | Recessive decoupling |
| 2013 | −2.60 | 4.84 | −0.54 | Strong decoupling |
| 2014 | −7.29 | 0.75 | −9.72 | Strong decoupling |
| 2015 | −6.85 | 7.42 | −0.92 | Strong decoupling |

growth rate was negative. These scenarios show that, although the textile industry has not fully realised the production of zero pollution in the water environment, the trend of decoupling between the two is increasing. From 2013 to 2015, the grey water footprint reduced in absolute terms, whilst the economy is growing, resulting in a strong decoupling.

### Double decoupling of water consumption and wastewater discharge

Table 5 exhibits the decoupling elasticity calculation results of water footprint and economic growth of China's textile industry from 2002 to 2015.

Data Sources: %ΔWF is the growth rate of water footprint; %ΔG is the economic growth rate; and $D_{G–WF}$ is the decoupling elasticity of water footprint and economic growth.

From the overall decoupling trend, the decoupling trend between China's textile industry water footprint and economic growth is good. From the change of decoupling states, it can be divided into three stages.

From 2002 to 2004, the decoupling between water footprint and economic growth is unstable, alternating between expansive coupling and strong decoupling. From 2005 to 2011, the decoupling of water footprint and economic growth improved. Although the development of China's textile industry has not yet gained independence from the water footprint, the negative impact of economic development on water environment decreased. From 2012 to 2015, strong decoupling and negative decoupling alternated between the water footprint and the industry's economy. For the first time, the growth rate of the water footprint in 2014 exceeded the growth rate of total output value of the textile industry. The elasticity coefficient was 5.28, showing a state of negative expansion decoupling, and the decoupling status was not optimistic. Given the change of the water footprint, this is due to the rapid growth of water resources consumption in textile industry in 2014, as well as the rebound of economic development, water resources consumption and water environmental pollution.

**Table 5 Decoupling elasticity of water resources environment and economic growth in China's textile industry from 2002 to 2015.**

| Year | %ΔWF | %ΔG | $D_{G-WF}$ | Degrees of decoupling/coupling |
|---|---|---|---|---|
| 2002 | 6.29 | 7.15 | 0.88 | Expansive coupling |
| 2003 | −8.61 | 7.66 | −1.12 | Strong decoupling |
| 2004 | 16.93 | 18.67 | 0.91 | Expansive coupling |
| 2005 | 11.18 | 22.86 | 0.49 | Weak decoupling |
| 2006 | −0.08 | 26.64 | −0.003 | Strong decoupling |
| 2007 | 7.79 | 18.85 | 0.41 | Weak decoupling |
| 2008 | −3.09 | 40.73 | −0.08 | Strong decoupling |
| 2009 | 1.09 | 6.38 | 0.17 | Weak decoupling |
| 2010 | 0.68 | 12.40 | 0.05 | Weak decoupling |
| 2011 | −9.71 | 25.39 | −0.38 | Strong decoupling |
| 2012 | −0.35 | −3.09 | 0.11 | Weak negative decoupling |
| 2013 | −2.58 | 4.84 | −0.53 | Strong decoupling |
| 2014 | 3.96 | 0.75 | 5.28 | Expansive negative decoupling |
| 2015 | −3.42 | 7.42 | −0.46 | Strong decoupling |

## Comparative study on the effectiveness of decoupling results

In the present study, the results of water consumption decoupling, wastewater discharge decoupling and the double decoupling of water consumption and wastewater discharge in China's textile industry from 2002 to 2015 are sorted and summarised. The effectiveness of the three decoupling types is compared with the absolute number and decoupling index of the decoupling years. On the whole, the double decoupling of water consumption and wastewater discharge in China's textile industry is the most unsatisfactory due to the simultaneous decoupling of water consumption and wastewater discharge impact. The increase of water resource consumption in a certain year or the increase of pollutant discharge in wastewater both lead to the deterioration of decoupling in that year.

### Absolute number comparison of decoupling years

Table 6 compares the results of water consumption decoupling, wastewater discharge decoupling and the double decoupling of water consumption and wastewater discharge in China's textile industry from 2002 to 2015.

In terms of the number of years to achieve decoupling, the decoupling of wastewater discharge reached 13 years, and only 2012 showed the recession decoupling. However, both water consumption decoupling and wastewater discharge decoupling indices failed to reach the decoupling index for 4 years. From the absolute number of strong decoupling years, the decoupling of water consumption is 7 years, the decoupling of wastewater discharge is 7 years, and the double decoupling of water consumption and wastewater discharge is only 6 years. In these 6 years, the absolute value of decoupling elasticity coefficient in 2 years is less than 0.1. Especially in 2006, the elasticity coefficient is −0.003, which has approached the critical value of strong decoupling and weak decoupling. In terms of the number of weak decoupling years, water consumption decoupled for

**Table 6 Absolute number comparison of decoupling years.**

| Water consumption decoupling | | | Wastewater discharge decoupling | | | Double decoupling of water consumption and wastewater discharge | | |
|---|---|---|---|---|---|---|---|---|
| Degrees of decoupling/coupling | Year | Elasticity coefficient | Degrees of decoupling/coupling | Year | Elasticity coefficient | Degrees of decoupling/coupling | Year | Elasticity coefficient |
| Strong decoupling | 2003 | −1.32 | Strong decoupling | 2002 | −0.84 | Strong decoupling | 2003 | −1.12 |
| Strong decoupling | 2006 | −0.1 | Strong decoupling | 2003 | −0.76 | Strong decoupling | 2006 | −0.003 |
| Strong decoupling | 2008 | −0.04 | Strong decoupling | 2008 | −0.14 | Strong decoupling | 2008 | −0.08 |
| Strong decoupling | 2009 | −0.05 | Strong decoupling | 2010 | −0.23 | Strong decoupling | 2011 | −0.38 |
| Strong decoupling | 2011 | −0.65 | Strong decoupling | 2013 | −0.54 | Strong decoupling | 2013 | −0.53 |
| Strong decoupling | 2013 | −0.53 | Strong decoupling | 2014 | −9.72 | Strong decoupling | 2015 | −0.46 |
| Strong decoupling | 2015 | −0.25 | Strong decoupling | 2015 | −0.92 | Weak decoupling | 2005 | 0.49 |
| Weak decoupling | 2005 | 0.66 | Weak decoupling | 2004 | 0.73 | Weak decoupling | 2007 | 0.41 |
| Weak decoupling | 2007 | 0.55 | Weak decoupling | 2005 | 0.18 | Weak decoupling | 2009 | 0.17 |
| Weak decoupling | 2010 | 0.2 | Weak decoupling | 2006 | 0.21 | Weak decoupling | 2010 | 0.05 |
| Expansive negative decoupling | 2002 | 1.99 | Weak decoupling | 2007 | 0.16 | Expansive coupling | 2002 | 0.88 |
| Expansive coupling | 2004 | 1.01 | Weak decoupling | 2009 | 0.65 | Expansive coupling | 2004 | 0.91 |
| Strong negative decoupling | 2012 | −0.69 | Weak decoupling | 2011 | 0.2 | Weak negative decoupling | 2012 | 0.11 |
| Expansive negative decoupling | 2014 | 13.45 | Recessive decoupling | 2012 | 1.48 | Expansive negative decoupling | 2014 | 5.27 |

3 years, wastewater discharge decoupled for 6 years, and finally double decoupling of water consumption and wastewater discharge occurred for 4 years.

### Decoupling index comparison

Each decoupling state is ranked and assigned to further quantify the performance of the three types of decoupling in the textile industry during the sample period. The decoupling index for evaluating the three types of decoupling state is then obtained. Figure 2 shows the decoupling index comparison of the three results.

In terms of the number of years to achieve strong decoupling, the total number of decoupling indices for strong decoupling years in water consumption decoupling is 184. The total number of decoupling indices for strong decoupling years in wastewater discharge decoupling is 190. The total number of decoupling indices for strong decoupling years in double decoupling of water consumption and wastewater discharge is only 158, which is much lower than the other two decoupling types. Combined with the situation of the weak decoupling years, the sum of the strong and the weak decoupling indices of the double decoupling of water consumption and wastewater discharge is the lowest, only 249. The sum of the strong and weak decoupling indices of water consumption decoupling and wastewater discharge decoupling reached 250 and 325, respectively. The result shows that despite the optimistic results on the number of years necessary for the textile industry to achieve double decoupling of water consumption and wastewater discharge, its decoupling index is not ideal within the decoupling range.
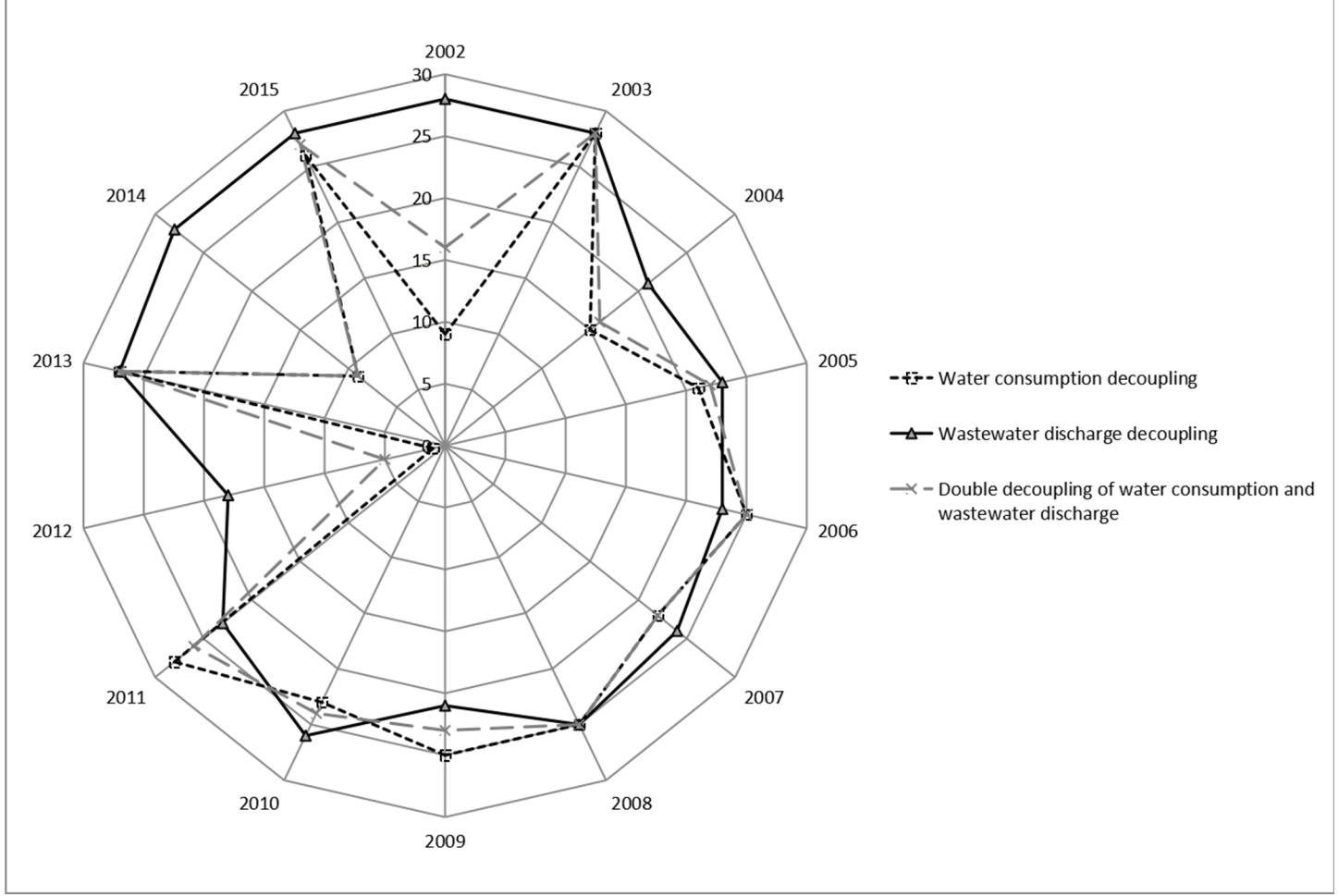

**Figure 2 Decoupling index comparison.** In terms of the number of years to achieve strong decoupling, the total number of decoupling indices for strong decoupling years in water consumption decoupling is 184. The total number of decoupling indices for strong decoupling years in wastewater discharge decoupling is 190. The total number of decoupling indices for strong decoupling years in double decoupling of water consumption and wastewater discharge is only 158, which is much lower than the other two decoupling types. Combined with the situation of the weak decoupling years, the sum of the strong and the weak decoupling indices of the double decoupling of water consumption and wastewater discharge is the lowest, only 249. The sum of the strong and weak decoupling indices of water consumption decoupling and wastewater discharge decoupling reached 250 and 325, respectively. The result shows that despite the optimistic results on the number of years necessary for the textile industry to achieve double decoupling of water consumption and wastewater discharge, its decoupling index is not ideal within the decoupling range.

According to the above analysis, whether from the perspective of absolute number of decoupling years or from the perspective of decoupling index, the requirement of double decoupling of water consumption and wastewater discharge is more stringent than the single water consumption decoupling or the wastewater discharge decoupling. One reason is that the double decoupling of water consumption and wastewater discharge computes for the economic system and the water resource consumption and wastewater discharge at the same time. Regardless of which aspect of the data is not ideal, decoupling will not occur. Water consumption and wastewater discharge require reduction to enhance the environmental effect of industrial water resources. Therefore, the double

decoupling of water consumption and wastewater discharge from the comprehensive perspective of resources and environment is of higher reference significance.

The comparison of decoupling effectiveness is based on the current situation of China's industrial water management. When formulating industrial water management policies, China lacks innovation of water management mode for the whole life cycle of industrial production and total water footprint control in industrial production process due to the over-detailed division of labor among departments or the existence of rights barriers. In the newer documents, the total amount of water intake is limited by the Regulations Governing Water Intake Permits and Collecyion of Water Resources Fees issued by the *State Council of the People's Republic of China (2017)*; the *Standing Committee of the National People's Congress (2017)* stipulates the total amount of wastewater discharge; and the Technical Guidelines for the Drafting of National Water Pollutant Discharge Standards (HJ 945.2-2018) issued by the *Ministry of Ecology and Environment of the People's Republic of China (2018)* directly affects industrial wastewater. The direct and indirect emission limits are specified. In 2013, although the *General Office of the State Council of China (2013)* strengthened the management at both ends. It stipulated the total water intake control target, water intake efficiency target and water quality control target, but there is still a lack of management means for the whole life cycle.

## CONCLUSIONS AND SUGGESTIONS

On the basis of the theory of water footprint decoupling, this paper makes a comparative analysis of water consumption decoupling, wastewater discharge decoupling and double decoupling of water consumption and wastewater discharge in China's textile industry and its sub-industries from 2001 to 2015.The main conclusions are as follows:

1. Water consumption decoupling and wastewater discharge decoupling in China's textile industry is higher. Amongst the 14 calculation results from 2002 to 2015, according to water consumption decoupling, 7 years show strong decoupling (2003, 2006, 2008, 2009, 2011, 2013, 2015), and 3 years show weak decoupling (2005, 2007, 2010). According to wastewater discharge decoupling, seven years show strong decoupling (2002, 2003, 2008, 2010, 2013, 2014, 2015) and 6 years show weak decoupling (2004, 2005, 2006, 2007, 2009, 2011). With the rapid expansion of China's textile economy, the progress of water saving and emission reduction technology and the higher level of enterprise management have improved the resource and environmental effects of industrial production.

2. The double decoupling of water consumption and wastewater discharge in China's textile industry shows alternating results although the overall decoupling status is good. The types of decoupling between economic growth of textile industry and water consumption and wastewater discharge in China include strong decoupling, weak decoupling, expansion coupling, weak negative decoupling and expansion negative decoupling. Amongst the 14 calculation results, 6 years show strong decoupling (2003, 2006, 2008, 2011, 2013, 2015), and 4 years show weak decoupling (2005, 2007, 2009, 2010). On the whole, the double decoupling of water consumption and wastewater

discharge of textile industry is higher, but the economic development of textile industry has not achieved complete independence from the water footprint.

3. The double decoupling of water consumption and wastewater discharge is more effective than single water consumption decoupling or single wastewater discharge decoupling. In terms of the number of years needed to achieve double decoupling of water consumption and wastewater discharge, 10 years showed double decoupling, with 6 years for strong decoupling and 4 years for weak decoupling. Compared with water consumption decoupling and wastewater discharge decoupling, the double decoupling result is lower. In terms of the decoupling index in the decoupling years, the sum obtained in the double-decoupling years was 249, lower than that in the high-decoupling years of water consumption (250) and wastewater discharge (325). Generally, the double decoupling of water consumption and wastewater discharge in China's textile industry is not optimistic compared with single water consumption decoupling and single wastewater discharge decoupling. This scenario verifies that the conditions for achieving the double decoupling of water consumption and wastewater discharge are more stringent, which means a higher reference significance for improving the management level of water resources.

The double decoupling of water consumption and wastewater discharge is an inevitable choice for the sustainable development of the textile industry. Promoting comprehensive water resource saving and recycling and enhancing the level of the comprehensive utilisation of water resources need green developments. The textile industry should carry out the life cycle water use management, issue the total water resources control and the total wastewater discharge control policy, and enforce the water footprint evaluation standard of the textile industrial park, textile enterprises and the textile products. A green industrial park, green products, green enterprises, optimised green supply chain, green institutional innovation and others could be comprehensively advanced. These measures could realise water metering type management innovation, improve the efficiency of structural water production, strengthen the effect of water management in enterprises, promote intensive water management ability and reduce the absolute amount for water and wastewater discharge. Thus, the textile industry could become the leading example of consumption reduction and water resource saving.

### Funding

This work was supported by the General Projects of Humanities and Social Sciences Planning of the Ministry of Education of China (19YJCZH092), the Zhejiang Provincial Natural Science Foundation of China (LY17G030035), the National Statistical Science Project, China (2018LY29) and the National College Students' Innovative Entrepreneurial Training Program of China (201810338044). The funders had no role in study design, data collection and analysis, decision to publish, or preparation of the manuscript.

## Grant Disclosures

The following grant information was disclosed by the authors:

General Projects of Humanities and Social Sciences Planning of the Ministry of Education of China: 19YJCZH092.

Zhejiang Provincial Natural Science Foundation of China: LY17G030035.

National Statistical Science Project, China: 2018LY29.

National College Students' Innovative Entrepreneurial Training Program of China: 201810338044.

## Competing Interests

The authors declare that they have no competing interests.

## Author Contributions

- Yi Li conceived and designed the experiments, performed the experiments, analysed the data, contributed reagents/materials/analysis tools, prepared figures and/or tables, authored or reviewed drafts of the paper, approved the final draft.
- Yi Wang performed the experiments, analysed the data, prepared figures and/or tables, approved the final draft.

## Data Availability

The raw data are provided in the Supplemental Files.

## Supplemental Information

Supplemental information for this article can be found online at http://dx.doi.org/10.7717/peerj.6937#supplemental-information.

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
