# Peer review of "Double decoupling effectiveness of water consumption and wastewater discharge in China’s textile industry based on water footprint theory"

_PeerJ, doi:10.7717/peerj.6937_

## Round 0.1 · original submission · Major Revisions

Based on the reviewers' comments, major revisions are needed.

Reviewer 1 ·

Basic reporting

no comment

Experimental design

no comment

Validity of the findings

no comment

Additional comments

•The way the authors analyze the double decoupling effectiveness of water consumption and wastewater discharge in China's
 textile industry is very specific. The results of this research can be used to constitute a practical tool in the realization of water decoupling in the textile industry.
•My only suggestion is to verify the English; minor errors were detected.

Reviewer 2 ·

Basic reporting

no comment

Experimental design

no comment

Validity of the findings

no comment

Additional comments

General Comments:

The paper mainly analyzes the water consumption decoupling, wastewater discharge decoupling, as well as the double decoupling of water consumption and wastewater discharge of China’s textile industry based on water footprint and decoupling theories. This work were conducted and analyzed extensively. However, some serious shortages must be revised.

Specific Remarks:
- This paper analyzed three sub-industries from 2001 to 2014. Why select those data in the years instead of the lasted data?

- Table 1. Emission limits for wastewater pollutants in the textile industry. Where did the data come from? Please give the cited papers or documents.

- Water Footprint Accounting Method had been reported. In this work, the method was used in textile industry by the authors. Can you illustrate the difference?

·

Basic reporting

This manuscript uses decoupling theory to study the relationship among the development of textile industry, resources consumption and environmental emissions. With the clear thoughts, appropriate approach and the credible results, this article provides a valuable reference to reduce resources consumption, decrease pollution emissions, and promote the green development of textile industry. Meanwhile, it will also make contribution to some relative research for other industries. However, there are still some shortcomings in this manuscript, which need to be further thought and improved.

The literature review needs to be modified. You may accept this following writing framework: 1) The current status of resources and environment research in China should be introduced in the first part (the research background). Then the situation of resources and environment in textile industry can be mentioned. In this part, you may briefly describe the current situation of Chinese textile industry, emphasis with the relationship changes between textile industry and water resources environment, and appropriately add some relevant studies on the textile industry and pollution emissions both in China and aboard. 2) The second part is about the decoupling theory and its application. In order to clarify the inherent meaning and application of decoupling theory, you need to focus on the judgement from decoupling index and to summarize the rule of industry development, then put forward some polices for future development.

Experimental design

The technical method of this paper is fitted, but based on outdated statistics. It will be better if you analysis with the past three years data from 2015 to 2017. The reason is that after 2015, China has vigorously promoted the construction of ecological civilization. Affected with taking ecological construction and industrial restructuring seriously, China's major pollutant emission has shown a downward trend.

Validity of the findings

The results basically match with the facts. But the reason analysis for decoupling or negative decoupling in textile industry needs be improved.

Countermeasures and suggestions are not deeply enough. You need to put forward some more pertinent suggestions, based on further analysis of the development rules and mechanism in textile industry in recent years.

Try to revise and improve relevant charts. For example, the results could be displayed in the form of radar maps to make the annual evolution of the research results more clearly.

Additional comments

A proof reading by a native English speaker should be conducted to improve both language and organization quality.

---

## Round 0.2 · accepted · Accept

Based on the reviewers' comments. The manuscript is acceptable.

Reviewer 2 ·

Basic reporting

no comment

Experimental design

no comment

Validity of the findings

no comment

Additional comments

The manuscript can be accepted.

·

Basic reporting

The manuscrip has improved the reference.

Experimental design

The experimental design is reasonable and the data has been updated.

Validity of the findings

The conclusion is correct. The suggestions will have reference to management.